# Audio Deepfake Detection with Self-Supervised XLS-R and SLS Classifier

Qishan Zhang
Hubei Minzu University
Enshi, China
zhangqishan2023@163.com

Shuangbing Wen
Hubei Minzu University
Enshi, China
202330267@hbmzu.edu.cn

Tao Hu*
Hubei Minzu University
Enshi, China
hutao_es@hbmzu.edu.cn

## Abstract

Generative AI technologies, including text-to-speech (TTS) and voice conversion (VC), frequently become indistinguishable from genuine samples, posing challenges for individuals in discerning between real and synthetic content. This indistinguishability undermines trust in media, and the arbitrary cloning of personal voice signals presents significant challenges to privacy and security. In the field of deepfake audio detection, the majority of models achieving higher detection accuracy currently employ self-supervised pre-trained models. However, with the ongoing development of deepfake audio generation algorithms, maintaining high discrimination accuracy against new algorithms grows more challenging. To enhance the sensitivity of deepfake audio features, we propose a deepfake audio detection model that incorporates an SLS (Sensitive Layer Selection) module. Specifically, utilizing the pre-trained XLS-R enables our model to extract diverse audio features from its various layers, each providing distinct discriminative information. Utilizing the SLS classifier, our model captures sensitive contextual information across different layer levels of audio features, effectively employing this information for fake audio detection. Experimental results show that our method achieves state-of-the-art (SOTA) performance on both the ASVspoof 2021 DF and In-the-Wild datasets, with a specific Equal Error Rate (EER) of 1.92% on the ASVspoof 2021 DF dataset and 7.46% on the In-the-Wild dataset. Codes and data can be found at https://github.com/QiShanZhang/SLSforADD.

## CCS Concepts

• **Security and privacy** → **Social aspects of security and privacy**; • **Information systems** → *Multimedia content creation*; • **Computing methodologies** → **Speech recognition**; • **Applied computing** → *Sound and music computing*.

## Keywords

Audio Deepfake Detection, Anti Spoofing, Countermeasures, Voice Conversion, AIGC, Text to Speech

---

*Corresponding author.

---

**ACM Reference Format:**
Qishan Zhang, Shuangbing Wen, and Tao Hu. 2024. Audio Deepfake Detection with Self-Supervised XLS-R and SLS Classifier. In *Proceedings of the 32nd ACM International Conference on Multimedia (MM '24), October 28-November 1, 2024, Melbourne, VIC, Australia* , 9 pages. https://doi.org/10.1145/3664647.3681345

## 1 Introduction

Deepfake audio produced by artificial intelligence algorithms like text-to-speech (TTS) [1], voice conversion (VC) [2] difficult to distinguish from real samples [3] have the potential to cause significant social and economic damage, have already been used to scam a CEO for 243.000$ [4]. Additionally, the potential for slander, misinformation, and fake news is enormous. There are also many biometric identity authentication applications, such as access control systems, telephone banking, and forensic scenarios [5]. Thus, there is a need for automatic detection and verification of human speech.

The audio deepfake detection task emerge as required by the times. The ASVspoof initiative and challenge series [6] was conceived to foster the development of countermeasures to protect against the manipulation of Automatic Speaker Verify (ASV) systems from spoofing attacks. The latest ASVspoof challenge is [7], which has introduced a new task: Deepfake (DF). This task aims to distinguish genuine utterances from AI-generated fake ones using machine learning techniques. The Audio deepfake detection challenge(ADD) [8, 9] has also held aim to fill the gap between the attack and the defense. Therefore, numerous deepfake audio detection algorithms have been proposed, which can essentially be classified into three categories: one based on handcrafted features such as Linear Frequency Cepstral Coefficients (LFCC) [10], Mel Frequency Cepstral Coefficients (MFCC) [11], Constant Q Cepstral Coefficients (CQCC) [12], etc., another on deep learning features like SincNet [13], FastAudio [14], and the third on features from pre-trained self-supervised models such as Wav2vec [15, 16], XLS-R [17], HuBERT [18], WavLM [19], and Whisper [20].

The flaw of handcrafted features is that they may overlook potentially useful characteristics for identifying deepfake audio, leading to deep learning features increasingly becoming the mainstream solution. Obtaining fake utterances for deep learning models to learn to extract features is costly and technically demanding, further complicated by the continuous emergence of new generative algorithms. Therefore, an efficient method involves using self-supervised pre-trained models that can be trained with any bona fide speech data. Using features from pre-trained models has achieved high detection accuracy. However, even with large pre-trained voice models, maintaining high accuracy against previously unseen neural network attack algorithms remains a challenge. Currently, the top-performing model on the ASVspoof 2021 deepfake (DF) eval dataset

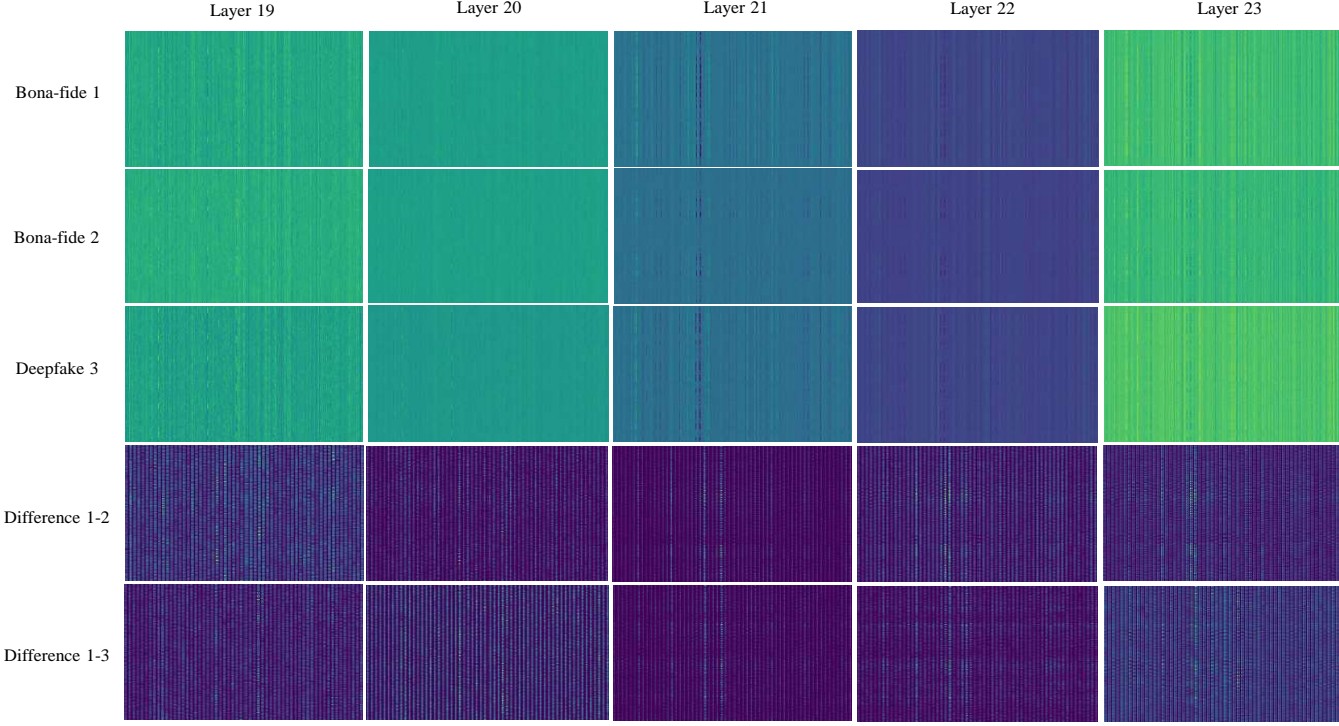

**Figure 1: XLS-R different layer show different level dicramenate feature.**

has achieved an EER of 2.56% [21], significantly lagging behind the 0.82% [16] achieved on the ASVspoof 2021 logical access (LA) eval dataset. The latter faces 17 unseen attacks, a number that is less than the array faced by the DF dataset.

In this paper, we are motivated by the following: **(i)** We believe that the features in the pre-trained, self-supervised XLS-R hidden layers contain useful discriminatory characteristics of audio, which can be very helpful in identifying deepfake audio, as shown in Figure 1. Regardless of whether the audio is real or deepfake, the features displayed by different hidden layers of XLS-R exhibit the same pattern. However, the difference map between real audios (row 4 of Figure 1) and the difference map between real and deepfake audios (row 5 of Figure 1) show different disparities in certain hidden layers. We believe that these differences may serve as effective features for distinguishing deepfake audio. **(ii)** When the hidden layers of XLS-R indeed offer more effective features for deepfake audio detection, is it still necessary to apply data augmentation and fine-tune pre-trained models?

The principal contributions of this work are:

- We propose a self-supervised, pre-trained XLS-R model-based SLS classifier for detecting deepfake audio. Features extracted from various XLS-R Transformer layers are fed into our classifier. The XLS-R features processed by the SLS module have a strong ability to distinguish between real and fake audio and exhibit strong generalization ability. The feature maps of randomly selected real and fake audio processed

by SLS have significant differences. At the same time, the module boasts rapid convergence and a simple architecture.
- For the first time, the method proposed in this paper has achieved an Equal Error Rate (EER) below 2% on the ASVspoof 2021 DF dataset, demonstrated competitive performance on the ASVspoof 2021 LA dataset, and secured state-of-the-art performance on the In-The-Wild dataset. Provides a new perspective for improving the robustness of deepfake audio detection models. Further analysis of experimental outcomes reveals that fine-tuning continues to play a pivotal role in enhancing the model's performance. Moreover, even with the exploitation of rich hidden layer information from the XLS-R model, the necessity for data augmentation persists.

## 2 Related Work

In Section 1, we introduce the most effective feature for audio deepfake detection when facing new generative algorithms: features from pre-trained models. This approach significantly enhances the model's recognition accuracy, thereby overcoming the bottleneck of limited training data availability. A number of pre-trained, self-supervised models are publicly available, including Wav2vec 2.0 [22], XLS-R [23] (a variant of Wav2vec 2.0), HuBERT [24], WavLM [19], and Whisper [20]. In this section, we briefly review prior work that utilizes pre-trained models to enhance model generalizability with the same limited training data as ours.

In 2021, Xie et al. [15] proposed utilizing features from the Wav2vec model, in conjunction with a Siamese neural network,

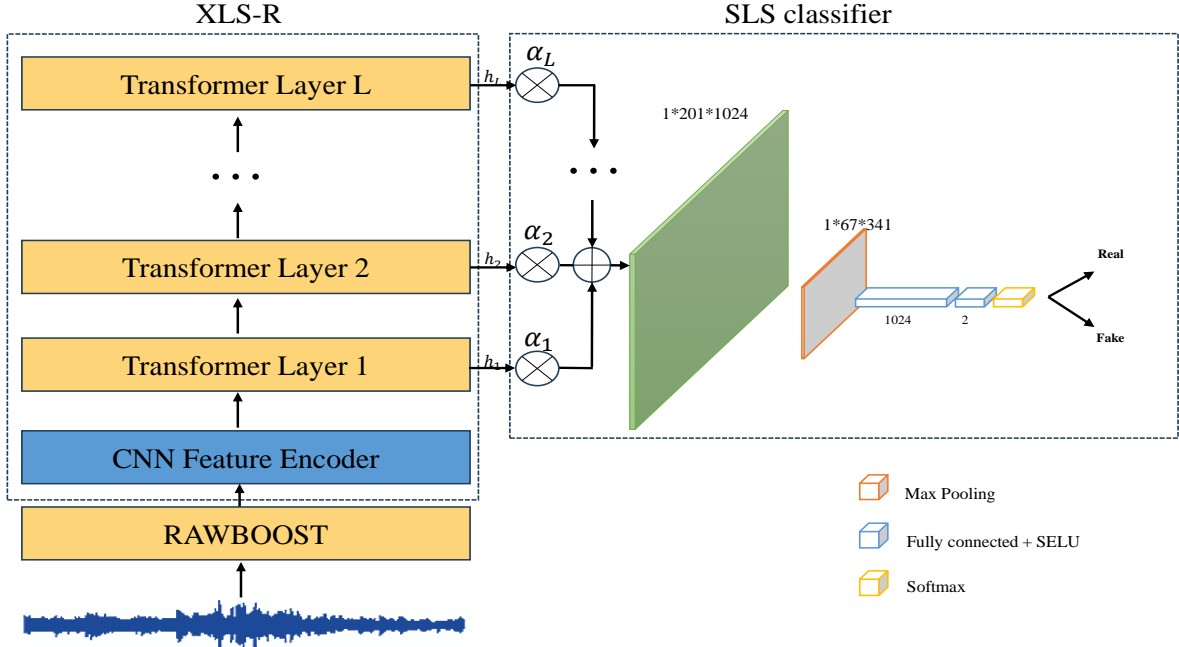

**Figure 2: Overview of the proposed audio deepfake detection approach base on XLS-R.**

for spoofing speech detection. This approach significantly reduced the Equal Error Rate (EER) from the previous state-of-the-art result of 4.07% to 1.15% on the ASVspoof 2019 evaluation set [25]. Martin-Donas [26] designed a system utilizing various transformer layers from XLS-128, a large-scale model for cross-lingual speech representation learning, paired with a simple downstream model to detect deepfake audio, achieving a 4.98% EER on the ASVspoof 2021 DF dataset. Wang et al. [18] explored various pre-trained, self-supervised speech models as the frontend for spoofing countermeasures. They experimented with Wav2vec2-small, Wav2vec2-large1, Wav2vec2-large2, HuBERT-XL, and XLS-53 (which is similar to Wav2vec2's structure but uses more training data). The experimental results suggest that the backend needs to be deep when the pre-trained frontend is not fine-tuned. In contrast, if the frontend can be fine-tuned, a simple backend with just average temporal pooling and a linear layer is sufficient. Additionally, the XLS-53 frontend with LGF classfier achieved a 4.75% EER on the DF dataset. Tak et al. [16] achieved significantly improved performance in the field of spoofing detection by applying the XLS-R model with a Rawnet2 [27] encoder and AASIST [28] backend, achieving an EER of 2.85% on the ASVspoof 2021 DF eval set. Furthermore, they demonstrated the use of data augmentation, showing its complementary benefits to self-supervised learning. Guo et al. [21] introduced the usage of WavLM as a frontend feature extractor and proposed the Multi-Fusion Attentive (MFA) classifier, based on the attentive statistics pooling layer. The MFA aggregates the output representations of WavLM, focusing on features at various layers and time steps, thus facilitating the extraction of highly discriminative features. These methods pushed the DF eval set's EER to 2.56%.

Wang et al. [29] utilized the pre-trained model HuBERT to extract duration features from the waveform, and employed a conformer to extract pronunciation features. These features were then fused with Wav2vec 2.0's output features via an attention mechanism, achieving a 29.53% EER on the In-the-wild dataset. Yang et al. [30] investigated the performance of a broad range of pre-trained models, including XLS-R, HuBERT, WavLM, and Whisper, coupled with a ResNet18 backend. They found that XLS-R performs best on the DF eval set, while HuBERT excels on the In-the-Wild dataset. They proposed two multi-view feature incorporation methods to capture the subtleties of multiple candidate features from XLS-R, WavLM, and HuBERT, thereby enhancing the system's performance and generalizability. This approach achieved a 24.27% EER on the In-the-Wild dataset.

## 3 Proposed Method

Our objective is to detect deepfake audio by classifying the contextualized representations derived from various transformer layer outputs of the pre-trained XLS-R model. Previous studies have demonstrated that, for many tasks such as speaker verification or emotion recognition, more discriminative information can be gleaned from the initial or intermediate layers of pre-trained models Since synthetic TTS algorithms cannot accurately mimic the real human speech flow and duration, the voice conversion process introduces vocoder artifacts into the audio signals, and some sound quality may be lost. This may result in the generated sound being unnatural or distorted. Pre-trained models can effectively capture these features for use as input in downstream models. Furthermore, the hidden layers of the transformer may provide representations more suitable for audio deepfake detection tasks. To achieve this,

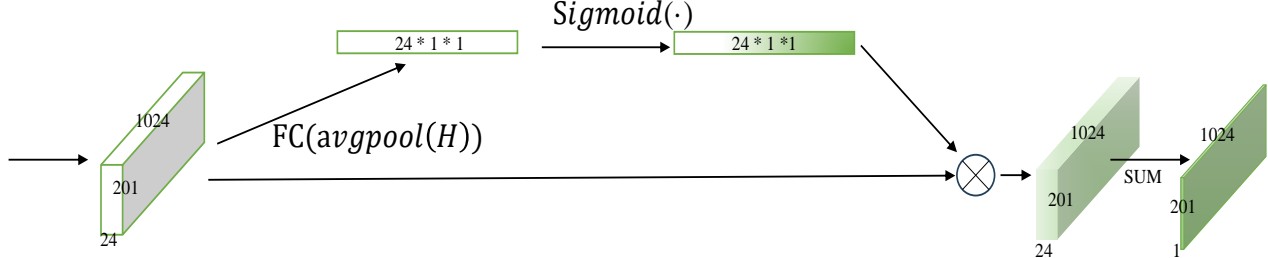

**Figure 3: The proposed Sensitive Layer Select(SLS) module.**

consider $x$ to be the waveform of a human voice signal labeled $y \in 0, 1$, where 0 signifies a real human voice and 1 denotes a synthetic human voice. Our aim is to construct a classifier $\hat{y} = F_\theta(x)$ to predict the label of input $x$. The binary detection model is constructed as a cascade of neural networks:

$$F_\theta(x) = C_{\theta_c}(W_{\theta_W}(x)) \tag{1}$$

where $W_{\theta_W}$ represents the front-end XLS-R model for audio representation, equipped with its own set of parameters $\theta_W$. $C_{\theta_c}$ denotes a backend binary classifier with a Sensitive Layer Select (SLS) module designed to select useful outputs from different XLS-R transformer layers, and $\theta_c$ are its parameters. This classifier can be optimized by solving:

$$\min_\theta \sum_{(x,y) \in T} \mathcal{L}_b(y, F_\theta(x)) \tag{2}$$

where $\mathcal{L}_b(y, \hat{y})$ represents the cross-entropy loss for binary classification, and $T$ denotes the training dataset comprising labeled real and synthetic examples.The model's overall framework is depicted in Fig. 2.Input raw waveforms are processed by the XLS-R model to obtain contextualized feature representations from various transformer layers.All contextualized representation features from the transformer layers are then input into our classifier.Within the classifier, features from various layers initially undergo weight adjustment, are subsequently summed, and then connected to a simple fully connected layer for binary classification.Detailed explanations of each component are presented in the subsequent subsections.

### 3.1 XLS-R model

The XLS-R [23] is a large-scale model for cross-lingual speech representation learning, based on wav2vec 2.0 and trained across 128 languages.The raw speech signal $x$ is initially processed by a feature encoder comprising several convolutional layers (CNN), extracting vector representations of size 1024 every 20ms, utilizing a receptive field of 25ms. This process yields $z$. Subsequently, these encoder features $z$ are input into a transformer network comprising 24 layers, which is utilized to derive contextualized representations $h$. The outputs from the distinct 24 transformer layers are concatenated as: $H = [h_1, h_2, ..., h_L]$. The model is trained in a self-supervised setting using a contrastive loss. The primary objective is to predict the quantized representations of specific masked encoded features from a set of distractors, utilizing the contextualized representations.Consequently, this model is capable of learning high-level

representations of the waveform signal. The features extracted from the pre-trained XLS-R model can be used to train a downstream classifier in a specific task with a relatively few amount of labeled data.Moreover, the XLS-R model and downstream models can be jointly trained in the related task. In this work, the utilization of XLS-R as a pre-trained model is explored. The mathematical representation of processing the audio signal $x$ through the XLS-R model is as follows:

$$H = W_{\theta_W}(x) \tag{3}$$

### 3.2 Classification Model

In this section, our objective is to employ algorithms capable of selecting useful contextualized representations from different XLS-R transformer layers. Inspired by the SENet, as proposed by Hu Jie et al. [31], it introduces a mechanism designed to recalibrate channel-wise feature responses by explicitly modeling interdependencies between channels. Given $H = [h_1, h_2, ..., h_L]$, our goal is to obtain:

$$F_\theta(x) = C_{\theta_c}(H) = \text{FC}(\text{maxpool}(\sum_{l=1}^{L} \alpha_l h_l)) \tag{4}$$

Where $h_l$ denotes the output of a distinct transformer layer in the XLS-R model, and $L$ represents the number of transformer layers.$\alpha_l$ represents the layer weight, defined as $\alpha = [\alpha_1, \alpha_2, ..., \alpha_{24}]$. This is derived from $H \in \mathbb{R}^{L \times N \times 1024}$, as illustrated in the equation below.

$$\alpha = \text{Sigmoid}(\text{FC}(\text{avgpool}(H))) \tag{5}$$

In the above equation, *avgpool* denotes an average pooling operation that is applied along the dimension $N$ of $H \in \mathbb{R}^{L \times N \times 1024}$. Subsequently, $\hat{H} \in \mathbb{R}^{L \times 1 \times 1024}$ is obtained, and through the full connected layer (FC), $\hat{H} \in \mathbb{R}^{L \times 1 \times 1}$ is produced. Our design rationale is that a single audio frame from avgpooling can represent the entire audio, and representing audio frames through a full connection can ascertain the layer's usefulness. A sigmoid function is then applied to these weights to ensure scaling between 0 and 1, thus facilitating dynamic channel-wise recalibration of the feature maps. Subsequently, the results of the 24 weighted layers are aggregated. This process is illustrated in Fig.3.

## 4 Experiment

To confirm the motivations outlined in the introduction, we conducted a series of experiments to evaluate the effectiveness of our proposed method. The first set entails jointly fine-tuning the XLS-R

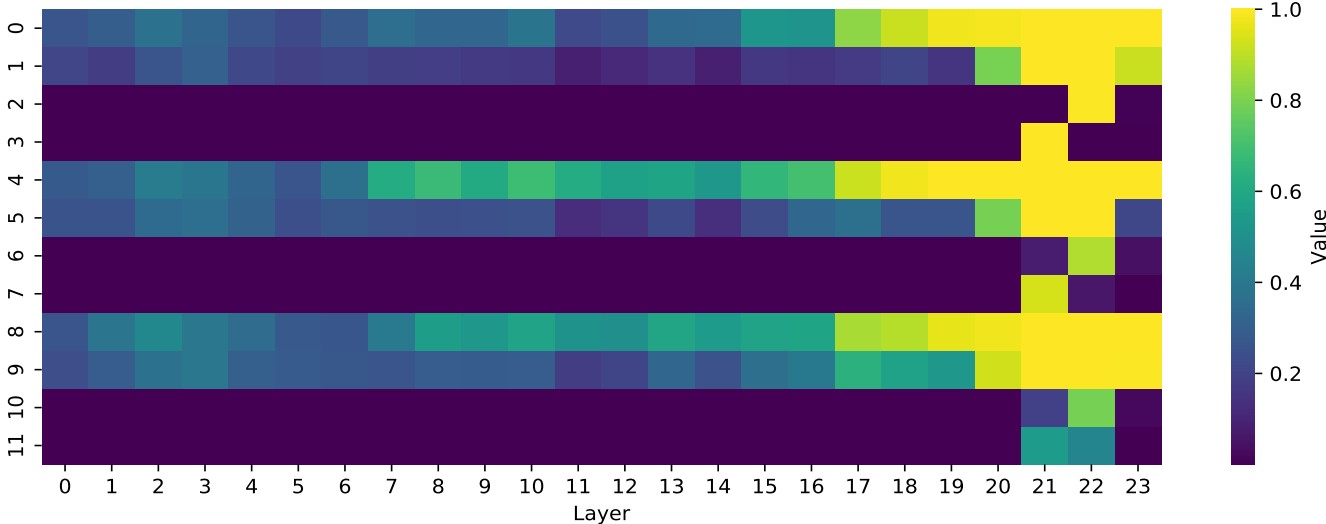

**Figure 4: Visualization of the weight values $\alpha$ of the SLS module demonstrates different attentions to various XLS-R features when dealing with real and deepfake audio. Where The first four rows display the attention weight distribution of the SLS module when processing audio in the DF dataset; rows 0 and 1 (0-1) respectively show the weight distribution for deepfake and real audio when handled by the SLS module; rows 2 and 3 (2-3) illustrate the weight distribution for fake and real audio respectively when the sigmoid function in the SLS module is replaced with softmax. Rows 4 to 7 show the attention weights for four different scenarios when processing audio in the LA dataset, and rows 8 to 11 for the In-the-Wild dataset.**

model with our proposed classifier and comparing its performance with those of other models on the ASVspoof 2021 LA (Logical Access), DF (DeepFake), and In-The-Wild evaluation datasets. Subsequently, we trained our model in scenarios without fine-tuning and without data augmentation. Finally, we conducted an ablation study to test our opinion that previous work limited the hidden layer's ability for audio deepfake detection. This section details the databases employed for training and evaluating our systems, alongside the data augmentation techniques and training setup procedures utilized. Then show the experiment result compartive with the state-of-the-art system, ans analyze the reason behind the result.

### 4.1 Datasets

We utilized the ASVspoof 2019 LA database [25, 32] training part for training, opting not to use the validation part, because we believe the validation set is traditionally used to prevent overfitting, but the validation part, compared to the train set, only differs in the speaker, we posit that the validation set can only prevent the model from overfitting to the train set's speaker features, it does not possess the ability to prevent overfitting produced by generative algorithms' features, and in the real world, it is necessary to confront features generated by various algorithms, thus, we exclusively use the train set and monitor training loss to determine the occurrence of overfitting. We evaluated our approach on the ASVspoof 2021 LA,DF [33], and In-The-Wild [34] evaluation sets. The 2021 LA dataset presents a challenge due to its inclusion of codec and transmission variability, elements not present in the training and

validation datasets. The DF dataset is more reflective of the performance of audio deepfake detection algorithms compared to the LA dataset; it contains over a hundred generation algorithms not seen in the training set, as well as compression variability during audio transmission. The In-The-Wild dataset consists of 37.9 hours of audio clips that are either fake (17.2 hours) or real (20.7 hours). The fake clips were created by segmenting 219 publicly available videos and audio files that explicitly advertise audio deepfakes. The corresponding genuine instances from the same speakers were collected using publicly available materials, such as podcasts, speeches, etc. The fake and genuine instances are similar in aspects such as background noise, emotions, and duration. Based on the above, the DF and In-the-Wild datasets are the most challenging and capable of reflecting the generality of models. The Equal Error Rate (EER) [35] is used as the evaluation metric.

**Table 1: The detailed information of the training sets, the development sets, ASVspoof2019 LA dataset and In-the-Wild dataset.**

| Set | Genuine | Spoofed | Total |
|---|---|---|---|
| | #utterance | #utterance | #utterance |
| Train | 2,580 | 22,800 | 25,380 |
| Eval (2021 LA) | 14,816 | 133,360 | 148,176 |
| Eval (2021 DF) | 14,869 | 519,059 | 533,928 |
| Eval (In-the-Wild) | 19,963 | 11,816 | 31,779 |

## 4.2 Training setup

Audio data is cropped or concatenated, yielding segments of approximately 4 seconds in duration (64,600 samples). We employ the pre-trained model XLS-R 300M provided by the literature [23], which is jointly optimised with the back-end SLS classifier using back-propagation [36], and utilize the Adam optimizer with the learning rate is $10^{-6}$ and weight_decay 0.0001. The batch size is 5. We set the training duration to 50 epochs, incorporating early stopping technology with a patience of 3. This indicates that when the training loss does not improve for three epochs, the training is stopped, and the model with the lowest loss is chosen for evaluation.

All models were trained on a single GeForce RTX 4090 GPU, and all results are reproducible using open source code[1] with the same random seed and GPU environment. RawBoost algorithm 3 [37] is used as the data augmentation method. The RawBoost algorithm adds white noise, which is used to simulate the confrontation caused by electromagnetic interference in audio spoofing detection. The white noise is processed by the FIR filter and added to the audio signal.

## 4.3 Comparison with State-of-The-Art

**Table 2: Comparative Pooled EER(%) results of our proposed method with other systems in the ASVspoof 2021 DF and LA evaluation set. Results are the best (average) obtained from three runs of each experiment with different random seeds.**

| model | DF | LA |
|---|---|---|
| CQCC & GMM [33] | 25.56 | 15.62 |
| LFCC & GMM [33] | 25.25 | 19.30 |
| LFCC & LCNN [33] | 23.48 | 9.26 |
| RawNet2 [33] | 22.38 | 9.50 |
| XLS-53 & LLGF [18] | 5.44 | 7.18 |
| XLS-R & FC & ASP [26] | 4.98 | 3.53 |
| XLS-53 & LGF [18] | 4.75 | 6.53 |
| XLS-R & Rawnet & ASSIST [16] | 2.85 | 4.11 |
| WavLM & MFA [21] | 2.56 | 5.08 |
| **Ours** | **1.92(2.09)** | **2.87(3.88)** |

**Table 3: Comparison with other anti-spoofing systems In-The-Wild evaluation set, reported in terms of EER(%).**

| System | EER(%) |
|---|---|
| RawGAT-ST [34] | 37.81 |
| Wav2vec,HuBERT,Conformer & attention [29] | 36.84 |
| XLS-R & Res2Net [3] | 36.62 |
| MPE & SENet [38] | 29.62 |
| Spec & POI-Forensics [39] | 25.14 |
| XLS-R,WavLM,Hubert & Fusion [30] | 24.27 |
| XLS-R & Rawnet & ASSIST [16] | 10.46 |
| **Ours** | **7.46(8.87)** |

[1]https://github.com/QiShanZhang/SLSforADD

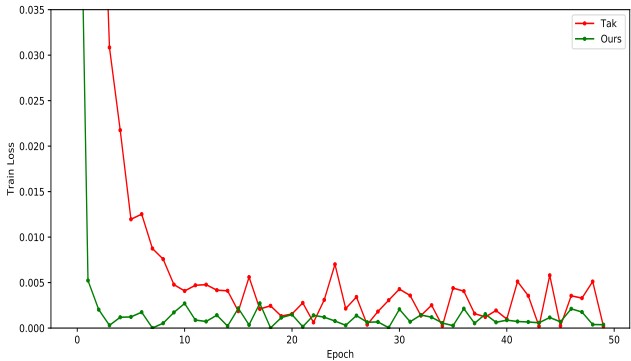

**Figure 5: Comparing the convergence speed of our model with the model proposed by Tak [16].**

Table 2 shows a comparison of our results with those of other models on the ASVspoof 2021 LA and DF evaluation datasets. Our method demonstrated the best performance on the DF dataset,. To our knowledge, this is the lowest reported equal error rate (EER) on the DF evaluation dataset, marking the first time EER has dropped below 2%. And achieve the lowest EER on the LA dataset. From Table 2, it can be seen that manual features such as CQCC and LFCC exhibit significant differences compared to pre trained features when combined with GMM or deep classifier LCNN. [26] and [21] also used hidden layer features, which proves the hypothesis we proposed at the beginning of the experiment. Previous models have limited the performance of hidden layer features, demonstrating the effectiveness of our proposed SLS classifier.

As shown in Table 4, audio generated by the Neural AR vocoder is the most difficult to distinguish under any condition. In the future, efforts can be made to explore methods to reduce this gap. It can be observed that our model has significantly minimized differences among various conditions; compared to the baseline condition C1, there are no significant disparities. Compared to Tak's model, which uses the same data augmentation and front-end, this demonstrates our SLS classifier's ability. The XLS-R features processed through our SLS module produce a very powerful feature map that can distinguish bona-fide or deepfake audio. This is illustrated in Figure 6.

Furthermore, in the LA dataset (Table 5), the transmission condition factors significantly influence, resulting in substantial disparities in EER. In Condition C1, which lacks transmission factors, algorithms such as A07, A08, A09, A13, and A14 generate deepfake audio that our model detects with 100% accuracy; moreover, different algorithms do not exhibit particularly large fluctuations in accuracy. The accuracy for Condition 6 dropped by approximately 700%. This is attributed to the use of Data Augmentation (DA) algorithms that adapt to data loss during the compression process. Our primary goal is to address the robustness of new generative algorithms; therefore, we did not implement data augmentation for transmission condition factors.

As demonstrated in Table 3, our model exhibited unprecedented generalization capabilities on the In-The-Wild dataset, achieving state-of-the-art performance. This result further substantiates the

**Table 4: Results in terms of Equal Error Rates (EERs, %) for each codec condition (DFC1-DFC9) and different generative algorithm vocoders on the ASVspoof 2021 DF evaluation partition. In the leftmost column, 'V' represents Vocoder, 'T' traditional vocoder, 'C' Wav concatenation, 'N' autoregressive neural vocoder, 'Nn' non-autoregressive use,'U' unkonw vocoder, and 'P' weighted pooling. In each condition, the left column represents our results, and the right column shows the results from Tak [16].**

| V | C1 | | C2 | | C3 | | C4 | | C5 | | C6 | | C7 | | C8 | | C9 | | Pooled | |
|---|---|---|---|---|---|---|---|---|---|---|---|---|---|---|---|---|---|---|---|---|
| T | 1.21 | 1.22 | 1.94 | 2.72 | 1.39 | 1.83 | 1.48 | 1.57 | 1.34 | 1.16 | 2.14 | 2.35 | 1.52 | 1.57 | 2.28 | 3.01 | 2.15 | 2.28 | 1.88 | 2.15 |
| C | 0.80 | 2.28 | 2.16 | 5.84 | 1.17 | 3.35 | 1.24 | 2.09 | 0.71 | 2.10 | 0.91 | 2.23 | 0.71 | 1.50 | 1.08 | 2.96 | 0.99 | 2.52 | 1.07 | 2.85 |
| N | 3.12 | 3.45 | 2.71 | 5.96 | 2.91 | 3.79 | 2.79 | 3.75 | 2.96 | 3.39 | 2.44 | 3.67 | 2.26 | 2.92 | 2.31 | 4.49 | 2.57 | 3.76 | 2.86 | 4.05 |
| Nn | 0.68 | 1.56 | 0.78 | 3.33 | 0.69 | 2.02 | 0.70 | 1.65 | 0.64 | 1.34 | 0.61 | 1.62 | 0.52 | 1.00 | 0.65 | 2.05 | 0.65 | 1.57 | 0.69 | 1.84 |
| U | 1.23 | 1.99 | 1.65 | 4.30 | 1.34 | 2.65 | 1.14 | 2.10 | 1.34 | 1.87 | 1.00 | 2.23 | 0.96 | 1.27 | 1.09 | 2.66 | 1.09 | 2.14 | 1.23 | 2.45 |
| P | 1.72 | 2.34 | 2.02 | 4.30 | 1.59 | 2.64 | 1.74 | 2.37 | 1.79 | 2.14 | 1.88 | 2.58 | 1.57 | 1.92 | 1.92 | 3.31 | 2.04 | 2.75 | **1.92** | 2.85 |

**Table 5: Results in terms of Equal Error Rates (EERs, %) for each transmission condition (LA C1-LA C9) and different generative algorithm on the ASVspoof 2021 LA evaluation partition.**

| algorithems | Input processor | Conversion | Wavform generator | C1 | C2 | C3 | C4 | C5 | C6 | C7 | Pooled |
|---|---|---|---|---|---|---|---|---|---|---|---|
| A07 | NLP | RNN* | WORLD | 0.00 | 0.22 | 0.15 | 0.15 | 0.13 | 0.18 | 0.18 | 0.18 |
| A08 | NLP | AR RNN* | Neural source-filter* | 0.00 | 0.36 | 0.43 | 0.30 | 0.20 | 0.55 | 0.37 | 0.41 |
| A09 | NLP | RNN* | Vocaine | 0.00 | 0.06 | 0.06 | 0.00 | 0.00 | 0.07 | 0.00 | 0.05 |
| A10 | CNN+bi-RNN* | ARRNN+CNN* | WaveRNN* | 0.88 | 1.98 | 3.19 | 1.61 | 1.83 | 2.33 | 3.10 | 5.93 |
| A11 | CNN+bi-RNN* | ARRNN+CNN* | Griffn-Lim | 0.91 | 1.41 | 2.37 | 1.35 | 1.46 | 1.48 | 3.48 | 4.79 |
| A12 | NLP | RNN* | WaveNet* | 0.30 | 0.83 | 0.58 | 0.76 | 0.80 | 0.55 | 1.36 | 1.76 |
| A13 | WORLD | Momentmatching* | Waveform filtering | 0.00 | 0.00 | 0.00 | 0.00 | 0.00 | 0.00 | 0.00 | 0.00 |
| A14 | ASR* | RNN* | STRAIGHT | 0.00 | 0.22 | 0.09 | 0.06 | 0.13 | 0.18 | 0.35 | 0.19 |
| A15 | ASR* | RNN* | WaveNet* | 0.06 | 0.61 | 0.36 | 0.67 | 0.50 | 0.50 | 0.48 | 0.73 |
| A16 | NLP | CART | Waveform concat. | 0.06 | 0.64 | 0.67 | 0.67 | 0.50 | 1.10 | 0.97 | 1.00 |
| A17 | WORLD | VAE* | Waveform filtering | 0.97 | 1.55 | 4.35 | 1.13 | 1.41 | 7.34 | 2.34 | 3.47 |
| A18 | MFCC/i-vector | Linear | MFCCvocoder | 1.27 | 1.72 | 5.32 | 1.44 | 1.74 | 5.70 | 2.42 | 3.63 |
| A19 | LPCC/MFCC | GMM-UBM | Spectral filtering+OLA | 0.88 | 1.50 | 4.74 | 1.19 | 1.28 | 8.27 | 2.41 | 3.69 |
| Pooled | | | | 0.51 | 1.15 | 2.38 | 1.08 | 1.10 | 3.48 | 2.16 | 2.87 |

effectiveness and generalizability of our proposed model in audio deepfake detection.

The underlying reason is likely attributable to our utilization of rich hidden layer features. Compared to prior work utilizing hidden layer features [21, 26], the weight assignment function is identified as playing a crucial role. These studies employ softmax for weight assignment, thereby limiting the classifier to the utilization of features from no more than one layer. The ablation study Section 4.4 and Figure 4 support this viewpoint.

Compared to prior work[16, 30], our model architecture achieves high accuracy despite its simplicity. This phenomenon prompts a reevaluation of audio deepfake detection model design principles. The comparison of model convergence speeds, as illustrated in Figure 5, shows our model converges faster than Tak's model[16], provided all other training hyperparameters remain constant.

This further substantiates the capability of XLS-R's hidden layers to provide a rich and effective feature set for discriminating against audio deepfake.

Based on the above experimental results, we introduce a new perspective on utilizing the XLS-R hidden layers, which contain rich features capable of enhancing deepfake audio detection performance. We propose an SLS classifier that uses a simple architecture

and achieves fast convergence, resulting in state-of-the-art (SOTA) performance across three challenging datasets.

### 4.4 Ablation Study

Demonstrated that every component of our proposed model is essential in this section. Table 6 presents the results of our ablation experiments on each component of the modified architecture. Clearly, the sensitive layer selection module plays a critical role in the model.

**Table 6: The ablation study intends to demonstrate the effectiveness of each part of the system.**

| Ablation | Configuration | DF | LA | In-The-Wild |
|---|---|---|---|---|
| ours | | **1.92** | 2.87 | **7.46** |
| w/o fine-tuning | XLS-R&SLS | 2.47 | **2.58** | 9.82 |
| w/o DA | XLS-R&SLS | 3.72 | 3.23 | 11.97 |
| w/o first five layer | XLS-R&SLS | 2.29 | 3.46 | 9.44 |
| w/o sigmoid | sum | 2.37 | 2.88 | 12.09 |
| | softmax | 3.01 | 3.41 | 8.89 |

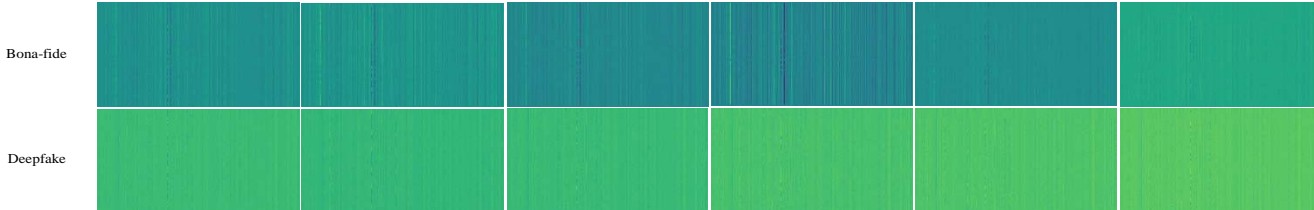

**Figure 6: Comparison of feature maps after the SLS module processes 6 bona fide audios and 6 deepfake audios randomly selected from the DF dataset.**

Clearly, we utilized the rich hidden layer features of XLS-R and the SLS effectively processed its features, our model still achieved cutting-edge performance without fine-tuning XLS-R. However, compared to fine-tuning, accuracy on the DF and In-the-Wild datasets declined. Conversely, on the LA dataset, there was an improvement, likely due to fine-tuning accelerating the model's adaptation to certain DF dataset features.

Without data augmentation, the model's performance significantly decreased across all three datasets, possibly due to conditions not encountered during XLS-R's pre-training, like encoding loss factors affecting audio data. In the absence of data augmentation, the model's accuracy was higher on the LA dataset than on the DF dataset, aligning with our intuition given the DF dataset's challenging nature. Accuracy also declined on the In-The-Wild dataset. This indicates that our data augmentation methods not only counteract encoding loss effectively but also accommodate the distribution of other real-world factors.

To test whether each hidden layer provides valuable discriminative features, we removed the first five layers of features with low weights, as shown in Figure 4. The experimental results showed that although the features in the first few layers were assigned lower weights, they still provided useful discriminative features. After removing these first five layers, the error rates of the three datasets increased.

The final ablation experiment confirmed our earlier hypothesis: replacing the weighted classification function in the SLS module with softmax limited the features provided by XLS-R. Simply aggregating all features from XLS-R yielded better results in the DF and LA datasets compared to softmax, whereas in the In-The-Wild dataset, softmax outperformed simple aggregation. We conclude that the In-The-Wild dataset encompasses the most diverse range of real-world factors affecting audio. Therefore, when XLS-R provides all feature representations, simply aggregating them results in chaotic feature maps from the SLS module, underscoring the need to differentiate between the features in real-world scenarios. The experiments suggest that our proposed SLS classifier is a promising option.

This contrasts with previous studies [26] that used softmax, a difference we attribute to our unique approach in feature processing. We treat weighted features as a single-channel image and apply direct pooling to reduce data volume. Unlike the fully connected layers employed in these studies, our pooling method retains more original features.

Upon further analysis of the SLS module's weight distribution map, we observed that the SLS module effectively discerns the disparity between the feature maps of genuine and fake audio. This selective aggregation of feature maps facilitates classification by downstream models. As illustrated in Figure 6, randomly selected examples from the DF dataset underscore this remarkable capability. Moreover, it exhibited consistency across various datasets. However, in rows eight and nine of the In-the-Wild dataset, the SLS module showed some blurring compared to its performance in the first two datasets, with less distinct attention to the differences between genuine and fake audio. By analyzing the weight distribution diagram of the SLS module in conjunction with the feature difference diagram of different audio samples from Figure 1, it can be observed that the primary differences between the gap diagram of real audio samples and the gap diagram between real and deepfake audio samples occur in layers 19, 20, and 23. This explains the phenomenon of weight distribution decreases for layers 19, 20, and 23 shown in Figure 6. However, this is just one possible preliminary explanation, and future work could involve more statistical efforts in the area of interpretability.

## 5 Conclusion

In this study, we validated the hypothesis that hidden layers of pretrained models harbor more abundant features for detecting audio deepfake. We introduced a classifier featuring a sensitive layer selection module, achieving state-of-the-art (SOTA) performance on two challenging datasets. One dataset, DF, features spoofed utterances generated by over 100 different attack algorithms, while In-The-Wild showcases real-world data distribution, thereby proving the approach's correctness and practical feasibility. Furthermore, our findings confirm that data augmentation is essential, even with the use of richer hidden layer features, and that fine-tuning enhances performance. While our model maintains state-of-the-art (SOTA) performance, it also boasts faster convergence and fewer parameters, prompting a reevaluation of future model design.

Future work could focus on designing strategies that more effectively utilize hidden layer features, as outlined in this paper, along with developing more precise and complex classification models. Our classifier structure is very simple, indicating significant potential for improvement. Utilizing more effective deep learning modules may offer further enhancements to model performance. The currently used pre-trained models are based on bonafide audio. A feasible solution might be to retrain these models with deepfake audio from more varied scenarios.

## 6 Acknowledgments

First and foremost, I would like to express my heartfelt gratitude to my Supervisor, Professor Tao Hu, for his invaluable guidance and support throughout the entire research process.

This work was supported by the Natural Science Foundation of Hubei Province of China under Grant 2023AFD061 and the Training Program of High-Level Scientific Research Achievements of Hubei Minzu University under Grant PY22011.

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
