# OpenReview forum: "Audio Deepfake Detection with Self-Supervised XLS-R and SLS Classifier"
_acmmm.org/ACMMM/2024/Conference — MM2024 Poster_

### Official Review · Reviewer_UUAs · 2024-05-14

**Rating:** 4
**Confidence:** 3

**Summary:**

The paper proposes a new deepfake audio detection model that leverages a pre-trained model called XLS-R and incorporates a Sensitive Layer Selection (SLS) module to enhance the detection of fake audio. The experiment shows that this method performs well on in-the-wild datasets.

**Strengths:**

1: This manuscript presents intriguing visual analyses, exemplified by Figures 1 and 4. The author's utilization of sigmod and softmax for result analysis is thought-provoking.

2: The proposed method demonstrates competitive performance in the wild dataset.

3: The proposed method exhibits simplicity and rapid convergence.

**Limitations:**

From a technical standpoint, the innovation contributed by this manuscript is limited when compared to [21].

**Suitability:**

2

---

### Official Review · Reviewer_qQAV · 2024-05-15

**Rating:** 2
**Confidence:** 3

**Summary:**

This paper makes use of the hidden layer features from XLS-R for audio deepfake detection. Experiments show that the proposed deepfake detection model achieves SOTA performance and the authors validate the effectiveness of the designed SLS module.

**Strengths:**

The paper is well-organized and the analysis of the method is detailed. The authors conduct numerous comparative experiments which make the proposed model to be persuasive.

**Limitations:**

-- about novelty
The main contribution of this paper is to use the hidden layer features of XLS-R for audio deepfake detection, in order to capture some useful information for detection. However, deepfake detection is essentially a binary classification task, and utilizing mid layer features for classification tasks is not a novel solution. What's more, the designed SLS module achieves the fusion of features taken from each layer, and such adaptive weight allocation is common in feature fusion.
I believe that the proposed model is going to achieve satisfactory results in ADD or ASVspoof challenges, but it lacks striking novelty as frontier research.

-- about experiment
How to prove that each hidden layer feature is effective and indispensable? The shallow hidden layer features which are fed to SLS for the fusion process have not been proven to be valuable in the experiment. As shown in Fig. 4, the weights of these layers are quite small compared to the deeper layers, and the connections between the Transformer layers result in a lot of redundancy in the underlying information contained in these mid layer features. For example, will excluding the hidden layer features of the first five Transformer layers cause a significant decrease in the model's deepfake detection performance? This is related to the core idea of the paper, but the paper failed to validate that.

-- others
In Fig. 3, the operations between data are missing and this figure seems to be unnecessary, as Fig. 2 also contains the detailed structure of the SLS module and I think Fig. 3 can be merged with Fig. 2.
The ablation study shows that fine-tuning and data augmentation significantly improve the model's performance. Has the fine-tuning and DA conducted for the baseline methods that utilize XLSR?

**Suitability:**

2

---

### Official Review · Reviewer_NDq5 · 2024-05-26

**Rating:** 4
**Confidence:** 3

**Summary:**

The authors proposed a deepfake speech detection method based on the combination of XLS-R and SLS techniques.

The proposed method was trained on ASVspoof19 dataset and evaluated on ASVspoof21 dataset and In-the-wild dataset.

The proposed method achieved reportedly the best performance on both datasets.

**Strengths:**

- The proposed method achieved reportedly the best performance on the popular ASVspoof21 and In-the-wild datasets
- The EER of the proposed method is significantly better than other methods on In-the-wild dataset

**Limitations:**

- The paper has a lot of formatting problems.
  - Please make sure there is a space between parenthesis and English text
  - Please make sure there is a space behind periods
  - Please make sure there is a space between citation bracket and English text

- In the abstract the authors refers to XLS-R without any further description, which is very confusing
- What is the difference between Layer Select Network (LSN) and Sensitive Layer Selection (SLS)?
- The description of XLS-R model is too vague. It is an important part of the proposed method, which requires more details.
- What are the pretrain data and method for XLS-R? Is there any potential overlap problem between the pretrain data of XLS-R and the test data used in this paper?
- What is the metric used in Table 4?
- Please briefly describe rawboost
- I am not convinced with claim the "human eye can directly identify the authenticity by observing the SLS feature maps" unless subjective tests are conducted
- The proposed method achieved significantly better performance compared to all other methods designed for in-the-wild dataset. The authors' explanation for this phenomenon is not very strong.
  - It seems like the authors proposed the SLS technique that boosted the performance significantly. The design of SLS seems to be applicable to other neural network architecture. Have the authors implemented the "rich feature" strategy for other neural network architectures? Does this strategy ubiquitously improve the performance of all neural network architectures? If so, the SLS technique itself seems to have more profound influence. If not, why does SLS work better with XLS-R?

**Suitability:**

3

---

### Official Review · Reviewer_bq28 · 2024-05-29

**Rating:** 4
**Confidence:** 4

**Summary:**

Thank you to all the authors for their efforts and for the opportunity to review the manuscript. In this paper, the authors propose a method for deepfake audio detection using pre-trained XLS-R transformer layer outputs and a sensitive layer selection module. Experiments show an improvement in deepfake audio detection performance on the ASVspoof 2021 DF Dataset and significant improvement on In-the-Wild Dataset. The code will be publicly released after paper’s acceptance.

**Strengths:**

1. Besides several typos, the introduction is well-written. The motivation and contributions of the paper are clearly highlighted, and help to get an overview of the entire paper.
2. The method’s novelty lies in the implementation of the combination of XLS-R transformer layer outputs and Layer Select Network.
3. It is good to see that XLS-R model provides good performance in feature selection as described in the paper. However, is there any reasoning behind choosing the cross-lingual XLS-R model and not any other pre-trained transformer model for speech recognition? Is there an ablation which supports this reasoning? If the answer is already present in the manuscript, please highlight this point carefully.
4. It is interesting to note the reasons behind not using a validation set to prevent overfitting because in this specific case, the validation set cannot fully prevent overfitting on the type of deepfake speech generation methods.
5. Exhaustive experimental analysis has been provided to prove the effectiveness of the proposed approach.

**Limitations:**

1. Line 94 - “Using features from pre-trained models has achieved high detection accuracy”. Is there a basis for this statement in the introduction? Perhaps, a citation or two can be added here to strengthen the argument. This is because any result comparing pre-trained models and hand-crafted features has not been provided in the paper or supplementary material. So, a citation will be helpful.
2. Figure 1, even though very illustrative, is not complete in analysis. This is because feature maps of only a single genuine audio and only a single deepfake audio are shown. What about the “Difference” between two genuine audios or two deepfake audios? If the difference is significant and perceptually identifiable in those cases too, then it invalidates the visual hypothesis related to feature maps. Please clarify.
3. Novelty: How is the proposed approach significantly different than a prior work “Martin-Donas[26]” in line 170, which also used transformer layer outputs and a downstream model, as described in the paper. Is the difference between the two approaches just the use of XLS-128 in one and XLS-R in the other? An elaboration about this in the relative work section would be helpful for the readers. Also, line 171 is confusing because it leaves the impression that XLS-128 is an entirely different model than XLS-R. Later in the paper, in line 332, it is mentioned that XLS-R is trained across 128 languages, which implies that both terms indeed refer to the same model. Similarly in Table 2, what is the difference between XLSR and XLS-R? So: 1) What separates the proposed method from prior work? 2) Is there a reason behind using different terms for “probably” the same model?
4. What features are “z” in Figure 2?
5. Table 2 and Table 3 look lighter w.r.t the number of comparison methods. For example, in Table 2, what is the performance of non XLS-R (e.g. Wav2vec) based models or spec based models? Since the methods in Table 2 and Table 3 do not overlap, it is difficult to conclude the correspondence in performance on both datasets.

Besides, there are some minor grammatical errors and typos which could be removed by proof-reading:

1. In Line 20, there should be “a” instead of “an” because SLS does not start with a vowel.
2. In Line 45, it should be “produced” instead of “produce”.
3. In Line 46, there is missing space between “text-to-speech” and “(TTS)”, similarly in “voice conversion” and “(VC)”. This has happened at several spaces in the paper. It needs to be made consistent.
4. Line 65-66, “There also …” has a missing “are”.
5. Line 68, “systems,telephone banking” has a missing space before ‘,’.
6. Line 82 - It is always proper to list full forms of abbreviations, the first time abbreviations are used in text, for example, LFCC, MFCC, EER and CQCC.
7. Figure 1 caption - Not sure what “dicramenate” is. The caption needs to be fixed grammatically.
8. Lines 196 and 197 -  XLSR has been listed twice within the same sentence, and is written differently than “XLS-R” in Line 151.
9. Line 267 -  End of sentence has a missing full stop.
10. Line 276 - it should be “{0,1}” as per set notation.

I am sure there are more which I could not list here. A proof-read and grammar check is strongly recommended. With the high amount of grammar errors and typos in the paper, some sections can become very hard to read.

Overall, the paper presents an interesting approach for deepfake audio detection. Some merits of the method include not requiring fine-tuning or augmentation, and direct extraction of distinguishing features from pre-trained transformer layers. There is a lot of experimental analysis provided in the paper, however comparison methods are not consistent. Moreover, there are numerous typos and grammar errors throughout the paper indicating a lack of proof-reading. More work is needed in the writing of the manuscript.

**Suitability:**

3

---

### Meta-Review · Area_Chair_AHSc · 2024-07-03

**Recommendation:** Accept (Poster)
**Confidence:** 4

**Metareview:**

The authors propose a deepfake speech detection method based on the combination of XLS-R and SLS techniques. The method is trained on the ASVspoof19 dataset and evaluated on the ASVspoof21 dataset and In-the-wild dataset. The paper reports good performance on both datasets.

Although the paper receives 'borderline accept' as the final decision from the reviewers, the rebuttal introduces new experiments and results. To me, for equitable evaluation against other conference submissions, I lean towards primarily considering the paper's existing results and experiments. The reviewer 'bq28' requests the inclusion of new content to the manuscript presented in the rebuttal. One reviewer ('NDq5' ) thinks that without a subjective experiment, the claims in the paper cannot be substantiated. However, in the rebuttal, it is stated that the subjective experiment was not conducted. It is recommended that the authors adequately address reviewer concerns in the camera ready version.

The reviewers highlight the following strengths and limitations:

Strengths:
1. The proposed method reports better performance on the popular ASVspoof21 and In-the-wild datasets

Limitations:
1. The subjective experiment was not conducted.
2. The suitability of Unimedia/unimodal in nature but of sufficient interest to the MM community
3. The innovation contributed by this manuscript is limited when compared to [21]
4. The grammar and typos.